# Gene Regulation by Antitumor *miR-204-5p* in Pancreatic Ductal Adenocarcinoma: The Clinical Significance of Direct RACGAP1 Regulation

**DOI:** 10.3390/cancers11030327

**Published:** 2019-03-07

**Authors:** Muhammad Khalid, Tetsuya Idichi, Naohiko Seki, Masumi Wada, Yasutaka Yamada, Haruhi Fukuhisa, Hiroko Toda, Yoshiaki Kita, Yota Kawasaki, Kiyonori Tanoue, Hiroshi Kurahara, Yuko Mataki, Kosei Maemura, Shoji Natsugoe

**Affiliations:** 1Department of Digestive Surgery, Breast and Thyroid Surgery, Graduate School of Medical Sciences, Kagoshima University, Kagoshima 890-8580, Japan; stars819@gmail.com (M.K.); idichitetuya@gmail.com (T.I.); k8911571@kadai.jp (M.W.); k8771744@kadai.jp (H.F.); nefeltari2000@yahoo.co.jp (H.T.); north-y@m.kufm.kagoshima-u.ac.jp (Y.Ki.); gekayota@gmail.com (Y.Ka.); wilson@m.kufm.kagoshima-u.ac.jp (K.T.); h-krhr@m3.kufm.kagoshima-u.ac.jp (H.K.); mataki@m.kufm.kagoshima-u.ac.jp (Y.M.); kmaemura@m3.kufm.kagoshima-u.ac.jp (K.M.); natsugoe@m2.kufm.kagoshima-u.ac.jp (S.N.); 2Department of Functional Genomics, Chiba University Graduate School of Medicine, Chiba 260-8670, Japan; yasutaka1205@olive.plala.or.jp

**Keywords:** microRNA, *miR-204-5p*, antitumor, pancreatic ductal adenocarcinoma, pathogenesis, *RACGAP1*

## Abstract

Previously, we established a microRNA (miRNA) expression signature in pancreatic ductal adenocarcinoma (PDAC) tissues using RNA sequencing and found significantly reduced expression of *miR-204-5p*. Here, we aimed to investigate the functional significance of *miR-204-5p* and to identify *miR-204-5p* target genes involved in PDAC pathogenesis. Cancer cell migration and invasion were significantly inhibited by ectopic expression of *miR-204-5p* in PDAC cells. Comprehensive gene expression analyses and *in silico* database searches revealed 25 putative targets regulated by *miR-204-5p* in PDAC cells. Among these target genes, high expression levels of *RACGAP1*, *DHRS9*, *AP1S3*, *FOXC1*, *PRP11*, *RHBDL2* and *MUC4* were significant predictors of a poor prognosis of patients with PDAC. In this study, we focused on *RACGAP1* (Rac guanosine triphosphatase-activating protein 1) because its expression was most significantly predictive of PDAC pathogenesis (overall survival rate: *p* = 0.0000548; disease-free survival rate: *p* = 0.0014). Overexpression of *RACGAP1* was detected in PDAC clinical specimens, and its expression enhanced the migration and invasion of PDAC cells. Moreover, downstream genes affected by *RACGAP1* (e.g., *MMP28*, *CEP55*, *CDK1*, *ANLN* and *S100A14*) are involved in PDAC pathogenesis. Our strategy to identify antitumor miRNAs and their target genes will help elucidate the molecular pathogenesis of PDAC.

## 1. Introduction

Due to its aggressive nature, pancreatic ductal adenocarcinoma cancer (PDAC) is the deadliest of all solid cancers [1]. Patients with PDAC often present without symptoms and, by the time of diagnosis, exhibit lymph node and distant metastases as well as vessel invasion [2]. The prognosis of patients with PDAC has not been improved by surgery or current chemotherapy, with the 5-year survival rate remaining at only 5−7% [3]. Recently, molecular target therapies for several cancers have shown remarkable therapeutic efficacy; however, no targeted therapeutics are currently approved for treatment of PDAC [4]. Previous studies have demonstrated that KRAS, TP53 and SMAD are driver oncogenes in PDAC [5]. To improve prognosis, further studies are required to identify anticancer molecules. Comprehensive analysis of RNA networks in PDAC cells is indispensable for the development of novel therapeutic strategies for lethal PDAC.

Developing high-throughput genotyping technologies have been accelerated personalized medicine. Through the latest Genome-Wide Association Studies (GWAS) assays, several genetic risk variants involved in the risk of PDAC developing have been detected [6,7,8,9]. Importantly, GWAS catalogue shows that functional genetic variants revealed by GWAS were localized in non-coding region of human genome. In the near future, GWAS analyses may show variants of miRNAs and risk of developing PDAC.

MicroRNAs (miRNAs) are small non-coding RNAs (19–22 nucleotides, single stranded) that act to fine-tune the expression of protein-coding and non-coding RNAs in a sequence-specific manner [10,11,12,13,14,15]. As a unique characteristic of miRNAs, a single miRNA regulates a vast number of RNA transcripts within a cell [10,11,12,13,14,15]. Therefore, it is possible to identify novel RNA networks based on miRNA regulation using the latest genome analysis strategies in a given cell type. Accumulating evidence has shown that aberrantly expressed miRNAs actually act as oncogenes or tumor suppressors in human cancer cells and are involved in cancer pathogenesis [10,11,12,13,14,15].

In our continuing analyses of PDAC pathogenesis, we attempted to identify antitumor miRNAs and the cancer pathways that they control. Recently, we successfully created the miRNA expression signature of PDAC using RNA-sequencing technologies [16]. Based on antitumor miRNAs by miRNA signature, we have sequentially demonstrated novel RNA networks in PDAC cells [17,18,19,20]. Our previous studies showed significant downregulation of five miRNAs clustered on chromosome 2p16.1 (*miR-217*, *miR-216a-5p*, *miR-216a-3p*, *miR-216b-5p* and *miR-216b-3p*) in PDAC tissues, and ectopic expression of these miRNAs inhibited cancer cell migration and invasion [16,17]. A search of oncogenes targeted by miRNAs revealed that *miR-216b-3p* and *miR-217* directly regulate Forkhead box Q1 (*FOXQ1*) and actin-binding protein Anillin (*ANLN*) in PDAC cells, respectively [16,17].In addition, *FOXQ1* and *ANLN* were found to be overexpressed in PDAC clinical specimens and acted as oncogenes in PDAC cells [16,17].Searching for novel expression networks based on miRNAs is an effective strategy to identify molecular targets of PDAC.

RNA-sequencing analyses of miRNA expression signatures revealed that *miR-204-5p* was significantly downregulated in PDAC tissues. Although a tumor suppressor function of *miR-204-5p* has been reported in several cancers, *miR-204-5p* regulation of RNA networks in PDAC is still obscure. Here, we aimed to investigate the antitumor roles of *miR-204-5p* and to identify *miR-204-5p*-regulated oncogenes involved in PDAC pathogenesis. Comprehensive gene expression analyses and *in silico* database searches revealed that 25 putative targets are regulated by *miR-204-5p* in PDAC cells. Among these targets, high expression of seven genes (*RACGAP1*, *DHRS9*, *AP1S3*, *FOXC1*, *PRP11*, *RHBDL2* and *MUC4*) was significantly associated with a poor prognosis of patients with PDAC according to analyses of The Cancer Genome Atlas (TCGA) database. In this study, we focused on *RACGAP1* (Rac GTPase-activating protein 1) and performed further cell functional analyses. Our present data may provide new insights into the potential mechanisms of PDAC aggressiveness.

## 2. Results

### 2.1. Expression of miR-204-5p in PDAC Specimens and Cell Lines

Expression of *miR-204-5p* was evaluated in normal pancreatic tissues (*n* = 17), PDAC tissues (*n* = 24) and PDAC cell lines (SW1990 and PANC-1). Expression of *miR-204-5p* was found to be significantly downregulated in the PDAC specimens and both cell lines than in normal pancreatic tissues (Figure 1A). The clinical features of the patients with PDAC and the normal tissues are summarized in Table 1; Table 2, respectively.

### 2.2. Effects of Overexpressing miR-204-5p on the Proliferation, Migration and Invasion of PDAC Cells

To evaluate the antitumor effects of *miR-204-5p*, we applied gain-of-function assays (mature miRNAs transfection) in PDAC cell lines (SW1990 and PANC-1). PDAC cell proliferation was not affected by *miR-204-5p* overexpression (Figure 1B). In contrast, ectopic expression of *miR-204-5p* significantly prevented PDAC cell migration and invasion (Figure 1C,D).

### 2.3. Identification of Putative Genes Regulated by miR-204-5p in PDAC Cells and Their Clinical Significance

To identify putative oncogenes regulated by *miR-204-5p* in PDAC cells, in silico analyses were combined with the results of our genome-wide gene expression analyses using an oligonucleotide microarray (in SW1990 cells transfected with *miR-204-5p*; Gene Expression Omnibus (GEO) accession number: GSE115801). Our strategy to identify putative target genes regulated by *miR-204-5p* in PDAC cells is shown in Appendix A. From these analyses, 25 genes were identified as candidate oncogenes regulated by *miR-204-5p* in PDAC (Table 3).

Next, to investigate the clinical significance of these target genes, we determined the relationships between gene expression levels and prognosis (i.e., OS (overall survival) and DFS (disease-free survival) rates) in patients with PDAC using data from TCGA database. Significant associations were detected between upregulated expression of seven genes (*RACGAP1, DHRS9, AP1S3, FOXC1, RHBDL2, MUC4* and *PRR11*) and a poor prognosis in patients with PDAC (OS: *p* < 0.05) (Table 3 and Figure 2).

We also investigated the pathway analyses using *miR-204-5p* controlled downregulated genes. Several pathways were identified as *miR-204-5p* controlled pathways, e.g., “Regulation of actin cytoskeleton”, “Cytokine-cytokine receptor interaction”, “MAPK signaling pathway” and “Endocytosis” (Appendix A). In this study, we focused on *RACGAP1* because its expression level was most significantly associated with a poor prognosis in patients with PDAC (OS: *p* = 0.0000548; DFS: *p* = 0.0014) (Figure 2).

### 2.4. Expression of RACGAP1 in PDAC Clinical Specimens and Its Clinical Significance

The mRNA expression level of *RACGAP1* was significantly upregulated in PDAC tissues (Figure 3A). Negative correlations between *RACGAP1* mRNA expression and *miR-204-5p* expression were analyzed by Spearman’s rank test (R = −0.381, *p* < 0.0070, Appendix A). We extracted samples of PDAC from the TCGA database. We analyzed clinicopathological factors of *miR-204-5p* and *RACGAP1* expression (*miR-204-5p*; Appendix A, *RACGAP1*). The recurrence of *RACGAP1* showed high expression with a significant difference (*p* < 0.0015). Immunostaining revealed expression of the RACGAP1 protein in PDAC lesions but a lack of expression in noncancerous epithelial tissues (Figure 3C).

Kaplan-Meier analyses showed that OS and DFS rates were significantly lower in patients with PDAC exhibiting elevated *RACGAP1* expression compared with low expression (Figure 2). The clinical significance of *RACGAP1* expression in terms of OS in patients with PDAC was further assessed by univariate and multivariate Cox hazard regression analyses. *RACGAP1* expression was an independent predictor of OS according to the multivariate analysis (hazard ratio = 2.266, *p* = 0.0002; Figure 3B).

In addition, TCGA database analyses showed that high expression of *RACGAP1* was closely associated with poor prognosis (3-year overall survival) of several cancers, e.g., kidney renal papillary cell carcinoma, hepatocellular carcinoma, lung adenocarcinoma, sarcoma and skin cutaneous melanoma (Appendix A).

### 2.5. Direct Regulation of RACGAP1 by miR-204-5p in PDAC Cells

We then assessed whether *miR-204-5p* regulates the expression of *RACGAP1* in PDAC cells. We first confirmed significant downregulation of *RACGAP1* mRNA (Figure 4A) and protein (Figure 4B) levels in PDAC cells (SW1990 and PANC-1) transfected with *miR-204-5p* by qRT-PCR and western blotting, respectively.

According to the TargetScanHuman 7.2 database, *RACGAP1* harbors one binding site for *miR-204-5p* at nucleotide positions 851-857 in the 3′-UTR (Figure 4C, upper).

Next, we performed dual luciferase reporter assays to determine whether this site within *RACGAP1* is directly targeted by *miR-204-5p* by transfecting cells with plasmids harboring wild-type *RACGAP1* or *RACGAP1* with deletion of positions 851–857 in the 3′-UTR. *miR-204-5p* suppressed luciferase reporter activity in SW1990 and PANC-1 cells transfected with the wild-type *RACGAP1* vector compared with the mock- or miR-control-transfected cells. On the other hand, luciferase reporter activity was not decreased in cells transfected with the vector harboring the *RACGAP1* 3′-UTR deletion (Figure 4C, lower). These results suggest that *miR-204-5p* directly binds to putative binding site within the 3′-UTR of *RACGAP1*.

### 2.6. Effects of RACGAP1 Knockdown on the Proliferation, Migration and Invasion of PDAC Cells

To assess the function of *RACGAP1* in PDAC cells, loss-of-function assays using siRNAs targeting *RACGAP1* (si-*RACGAP1*) were performed. High expression levels of *RACGAP1* in two cell lines, SW1990 and PANC-1, were confirmed by PCR (Figure 3A). We evaluated the knockdown efficiency of RACGAP1 in si-*RACGAP1*-transfected SW1990 and PANC-1 cells and confirmed downregulation of RACGAP1 mRNA and protein levels (Figure 5A,B).

Cancer cell migration and invasion were significantly inhibited in si-*RACGAP1*-transfected cells compared with mock- or *siRNA*-control-transfected PDAC cells. However, transfection of si-*RACGAP1* did not significantly affect cell proliferation (Figure 5C–E).

EMT-related genes were selected and picked up from gene expression analysis data (GSE115801 and GSE115909). Our array data showed that expression of *SNAL3* and *vimentin* were reduced by si-*RACGAP1* transfection into SW1990 cells. Expression of fibronectin 1 was reduced by *miR-204-5p* transfection into SW1990 cells (Appendix A).

### 2.7. Identification of RACGAP1-Regulated Genes in PDAC Cells

To identify genes affected downstream of *RACGAP1*, we applied genome-wide gene expression analyses using SW1990 cells transfected with si-*RACGAP1*. The gene expression data were deposited in the GEO database (GEO accession number; GSE115909). Our strategy to identify *RACGAP1*-regulated genes is shown in Appendix A. Using this strategy, we identified 64 putative genes affected by *RACGAP1* in PDAC cells (Table 4). Among these genes, high expression levels of 12 genes (*MMP28, CEP55, CDK1, ANLN, S100A14, SLC6A14, TRIM29, TMPRSS4, SERPINB3, CAPN8, MELK* and *FAR2*) were significantly associated with overall survival in patients with PDAC, according to analyses of data from TCGA database (Figure 6).

Furthermore, we provided a heatmap gene visualization and validated as a prognostic ability of these 12 genes (Figure 7). As shown in Figure 7, patients with high gene signature expressions (Z-score > 0) were significantly poor OS and DFS rate than those with low gene signature expressions (Z-score ≤ 0) (OS; *p* = 0.0017, DFS; *p* = 0.0096, Figure 7). To explore the molecular network controlled by *RACGAP1* in PDAC cells, we performed pathway analyses using *RACGAP1* mediated downregulated genes (Appendix A).

Clinical significance of the expression levels of 12 genes (*MMP28*, *CEP55*, *CDK1*, *ANLN*, *S100A14*, *SLC6A14*, *TRIM29*, *TMPRSS4*, *SERPINB3*, *CAPN8*, *MELK* and *FAR2*) based on data from The Cancer Genome Atlas database. Kaplan-Meier plots of overall survival with log-rank tests comparing the survival of PDAC patients with high (red lines) versus low (blue lines) expression levels of each of the abovementioned genes.

## 3. Discussion

Numerous protein-coding and noncoding RNAs can be regulated by a single miRNA. Altered miRNA expression can disrupt RNA networks within cancer cells; therefore, manipulation of miRNA expression in cancer cells will help identify novel RNA networks with critical roles in cancer. To obtain novel therapeutic targets in PDAC, we sequentially identified antitumor miRNA-regulated oncogenes and pathways based on our original PDAC miRNA signature [16].

In this study, we focused on *miR-204-5p* and its regulated genes involved in PDAC pathogenesis. Previous studies indicated that *miR-204-5p* is downregulated in several cancers, and that restoration of *miR-204-5p* expression inhibits cancer cell proliferation, migration and invasion [21,22,23]. Our recent study showed greatest downregulation of *miR-204-5p* in triple-negative breast cancer (TNBC) tissues according to the miRNA signature of TNBC [24]. Ectopic expression of *miR-204-5p* in TNBC cells inhibited cancer cell migration and invasion, suggesting that *miR-204-5p* acts as an antitumor miRNA in these cells [24]. We reported that *miR-204-5p* is a pivotal antitumor miRNA preventing cancer cell progression. We are also interested in the RNA networks controlled by *miR-204-5p* according to cancer cell type.

Analyses of RNA networks regulated by *miR-204-5p* will contribute to our understanding of cancer pathogenesis. Our study identified a total of 25 genes as putative oncogenes by *miR-204-5p* regulation in PDAC cells. Among these targets, the high expression levels of seven genes (*RACGAP1*, *DHRS9*, *AP1S3*, *FOXC1*, *PRP11*, *RHBDL2* and *MUC4*) were significantly predictive of a poor prognosis in PDAC patients. These genes are highly involved in PDAC pathogenesis and are important landmarks for elucidating the molecular mechanisms underlying the aggressiveness of PDAC cells.

For example, overexpression of *FOXC1* (forkhead box C1), a member of the forkhead box of transcription factors, has been observed in several cancers [25]. Previous studies showed that aberrant expression of *FOXC1* enhances epithelial-to-mesenchymal transition and drug resistance [26,27]. In PDAC cells, *FOXC1* was found to be a pivotal regulator of insulin-like growth factor 1 receptor signaling, and aberrant activation of this signaling pathway contributed to PDAC cell aggressiveness, metastasis and an epithelial-to-mesenchymal transition phenotype [28]. These findings suggest that *FOXC1* is a potential therapeutic target for several cancers, including PDAC. Overexpression of *MUC4* (mucin 4), a transmembrane mucin, has been observed frequently in several cancers, including PDAC [29]. Mucins not only protect the epithelium from adverse conditions but also help transmit several signals into the cell [30]. The functions of *MUC4* are mediated by multiple receptors, and *MUC4* overexpression contributed to tumorigenesis, metastasis and drug resistance in PDAC cells [31]. Recently, we showed overexpression of *AP1S3* in TNBC specimens, and silencing of *AP1S3* inhibited the proliferation, migration and invasion of TNBC cells [24]. Previous studies showed that *FOXC1* was directly regulated by antitumor *miR-204-5p* in laryngeal squamous cell carcinoma and endometrial cancer [32]. Furthermore, *AP1S3* was directly regulated by *miR-204-5p* in TNBC cells [24]. The *miR-204-5p* target genes identified in our present study are highly involved in cancer pathogenesis.

We focused on *RACGAP1* because its elevated expression was most significantly predictive of a poor prognosis in PDAC patients (*p* = 0.0000548). Based on analysis of TCGA database, high expression of *RACGAP1* is significantly associated with breast cancer, lung adenocarcinoma, papillary renal cell carcinoma, hepatocellular carcinoma and cutaneous melanoma. These data suggest that aberrant expression of *RACGAP1* is involved in cancer pathogenesis. As a GTPase-activating protein, *RACGAP1* binds to activated G proteins to stimulate GTP hydrolysis, which in turn inactivates the G protein [33]. The function of GTPase-activating proteins opposes that of guanine nucleotide exchange factors, which stimulate G protein activation [34].

Our present data showed that *RACGAP1* was overexpressed in PDAC clinical specimens, and silencing of *RACGAP1* inhibited cancer cell migration and invasion. Our data suggest that *RACGAP1* acts as an oncoprotein rather than a tumor suppressor in PDAC cells. Recent studies reported overexpression of *RACGAP1* in several types of cancers (e.g., gastric cancer, colorectal cancer, uterine carcinosarcoma, hepatocellular carcinoma and epithelial ovarian cancer), and its aberrant expression was associated with poor prognosis in patients with these cancers [35,36,37,38,39]. In ovarian cancer, it was found that *RACGAP1* expression enhanced activation of RHOA and ERK proteins, and activation of its signaling induced cancer cell migration and invasion. In uterine carcinosarcoma cells, *RACGAP1* positively regulated STAT3 phosphorylation and survivin expression, and these activated signaling pathways induced an invasive phenotype [37]. In squamous cell carcinoma, *RACGAP1* was found to be regulated by the *E2F7* transcription factor, and its expression enhanced doxorubicin resistance [39]. These findings suggest that overexpression of *RACGAP1* is related to growth, progression, metastasis and drug resistance in various cancers. 

Finally, to investigate *RACGAP1*-regulated oncogenes and cancer-associated pathways in PDAC, we performed genome-wide expression analyses in PDAC cells after knockdown of *RACGAP1*. A total of 64 genes were identified as putative targets of *RACGAP1*. Among these targets, expression of 12 genes (*MMP28*, *CEP55*, *CDK1*, *ANLN*, *S100A14*, *SLC6A14*, *TRIM29*, *TMPRSS4*, *SERPINB3*, *CAPN8*, *MELK* and *FAR2*) was closely associated with a worse prognosis in patients with PDAC. Interestingly, aberrant expression of the actin-binding protein *ANLN* (anillin) was detected in PDAC clinical specimens, and knockdown of *ANLN* markedly inhibited the migration and invasion of PDAC cell lines [17]. In addition, that same study found that *ANLN* was directly controlled by antitumor *miR-217* in PDAC cells [17]. These results suggest that *RACGAP1* and *RACGAP1*-mediated genes are possible therapeutic targets for PDAC.

Interestingly, *TMPRSS4* and *SERPINB3* were identified as a PDAC classifier for discriminating PDAC and early precursor lesions from non-malignant tissue [40]. Notably, silencing of *TMPRESS4* expression blocked PDAC cell migration and invasion abilities [40]. Among these targets, we searched for regulation of miRNAs. Aberrant expression of the actin-binding protein *ANLN* (anillin) was detected in PDAC clinical specimens, and knockdown of *ANLN* markedly inhibited the migration and invasion of PDAC cell lines [17]. In addition, that same study found that *ANLN* was directly controlled by antitumor *miR-217* in PDAC cells. Ectopic expression of *miR-217* was significantly blocked cancer cell aggressiveness features [17]. Moreover, *TRIM29*, DNA and p53-binding protein is overexpressed in PDAC and its function as a mediator in DNA damage signaling [41]. Aberrant expression of *TRIM29* associated with radio-resistance in PDAC cells [41]. Tumor suppressive *miR-449a* inhibited PDAC cell aggressiveness through targeting *TRIM29* [41]. These results suggest that *RACGAP1* and *RACGAP1*-mediated genes are possible therapeutic targets for PDAC. Analysis strategy based on aberrant expressed miRNAs is effective for searching to novel molecular pathogenesis of PDAC.

## 4. Materials and Methods

### 4.1. Clinical Samples and PDAC Cell Lines

In this study, 24 PDAC clinical samples were collected from PDAC patients who underwent resection at Kagoshima University Hospital (Kagoshima, Japan) from 1997 to 2016. As controls, 17 pancreatic tissue specimens were collected from noncancerous regions. Gene expression analyses were conducted using total RNA extracted from cryopreserved PDAC tissues, and immunohistochemistry was performed using paraffin-embedded PDAC tissues. The clinical samples were staged according to the American Joint Committee on Cancer/Union Internationale Contre le Cancer TNM staging system. The clinicopathological factors of the clinical specimens are shown in Table 1. The present study was approved by the Institutional Review Board of Kagoshima University. Written informed consent and approval were provided by all patients (approval number: 16-27). 

We also used two PDAC cell lines in this study: SW1990 cells purchased from the American Type Culture Collection (Manassas, VA, USA) and PANC-1 cells purchased from RIKEN Cell Bank (Tsukuba, Ibaraki, Japan).

### 4.2. Quantitative Reverse-Transcription Polymerase Chain Reaction (qRT-PCR)

The procedure used for qRT-PCR has been described previously [42,43,44,45]. Reagents and equipment were purchased from Thermo Fisher Scientific (Waltham, MA, USA). TaqMan qRT-PCR probes targeting *miR-204-5p* (product ID: 000508) and *RACGAP1* (product ID: Hs01100049_mH) were obtained. Probes targeting *GUSB* (product ID: Hs99999908_m1) and *RNU48* (product ID: 001006) were used as internal controls. 

### 4.3. Transfection of Mature miRNAs and Small-Interfering RNAs (siRNAs)

The procedures for miRNA or siRNA transfection into cancer cells have been described previously [42,43,44,45]. The following mature miRNAs or siRNAs were used for transfection: *miR-204-5p* (product ID: PM11116), negative control miRNA (product ID: AM 17111) and two *RACGAP1* siRNAs (product IDs: HSS120934 and HSS120936). The oligonucleotides were transfected at 10 nM into PDAC cell lines using Lipofectamine RNAiMAX and Lipofectamine 2000 (Thermo Fisher Scientific, Waltham, MA, USA).

### 4.4. Cell Proliferation, Migration and Invasion Assays

As functional analyses, cell proliferation (XTT assay), migration (chamber cells migration assay) and invasion (Matrigel assay) assays were performed, as described previously [42,43,44,45].

### 4.5. Identification of Genes Regulated by miR-204-5p in PDAC Cells

To identify candidate target genes of *miR-204-5p*, comprehensive gene expression analyses using an oligonucleotide microarray (Agilent Technologies; Human Ge 60K, Santa Clara, CA, USA) were incorporated with *in silico* database analyses using the Target Scan Human 7.2 database (June 2016 release: http://www.targetscan.org/vert_71), as described previously (Appendix A) [42,43,44,45]. The microarray data were deposited into the Gene Expression Omnibus (GEO) repository (https://www.ncbi.nlm.nih.gov/ geo/). In addition, data from TCGA database regarding gene expression in PDAC were analyzed (https://cancergenome.nih.gov/).

### 4.6. Clinical Database Analyses of PDAC

TCGA database was used to investigate the clinical significance of PDAC miRNAs and the genes they regulated (https://tcga-data.nci.nih.gov/tcga/). Gene expression and clinical data were obtained from cBioPortal (http://www.cbioportal.org/) and OncoLnc (http://www.oncolnc.org) (data downloaded on 28 April 2018).

### 4.7. Plasmid Construction and Dual Luciferase Reporter Assay

The following 2 sequences were cloned into the psiCHECk-2 vector (C8021; Promega Corporation, Madison, WI, USA): the wild-type sequence of the 3′-untranslated regions (UTRs) of *RACGAP1*, or the deletion-type, which lacked the *miR-204-5p* target sites from *RACGAP1* (position 851–857). The procedures for transfection and dual-luciferase reporter assays were provided in previous studies [42,43,44,45]. 

### 4.8. Western Blotting and Immunohistochemistry 

For western blotting and immunostaining detection of RACGAP1 expression, anti-RACGAP1 antibodies (product ID: ab2270) were used according to the manufacturer’s instructions (Abcam Plc, Cambridge, UK). An anti-glyceraldehyde 3-phosphate dehydrogenase (GAPDH) antibody (product ID: SAF6698; Wako, Osaka, Japan) was used as an internal loading control for western blotting. Details of these methods have been described previously [42,43,44,45]. 

### 4.9. Downstream Genes Affected by RACGAP1 in PDAC Cells

Genome-wide gene expression and database oriented in silico analyses were applied to identify *RACGAP1*-mediated downstream genes. Our strategy for identification of genes regulated by *RACGAP1* is outlined in Appendix A. The microarray data used have been deposited in the GEO repository under accession number GSE115909.

### 4.10. Statistical Analysis

Comparisons between groups were assessed using the Mann-Whitney U test or one-way analysis of variance followed by Tukey’s multiple comparison test. Differences in survival rates were determined by Kaplan-Meier survival analysis and log-rank test. The Z-scores of target genes mRNA expression data and clinical sample information corresponding to PDAC patients were collected from cBioPortal. R2: Genomics Analysis and Visualization Platform (http://r2.amc.nl) was used to create a heatmap. Furthermore, Z-score was evaluated by a combination of each genes sets. High group (mRNA Z-score > 0) and low group (mRNA Z-score ≤ 0) were analyzed by Kaplan–Meier survival curves and log-rank statistics. The analyses were conducted using Expert StatView version 5.0 (SAS Institute, Inc., Cary, NC, USA) and GraphPad Prism version 7.04 (GraphPad Software, Inc., La Jolla, CA, USA).

## 5. Conclusions

In conclusion, *miR-204-5p* was downregulated in PDAC clinical specimens and acted as an antitumor miRNA by targeting several oncogenes involved in PDAC pathogenesis. *RACGAP1* was directly regulated by antitumor *miR-204-5p*, and high expression of *RACGAP1* significantly predicted a shorter survival in patients with PDAC. Overexpression of *RACGAP1* enhanced PDAC cell migration and invasion, suggesting *RACGAP1* as a possible therapeutic target for PDAC patients. Our approach to identify antitumor miRNAs and their regulated target genes in PDAC has potential value for the development of new therapeutic strategies.

## Figures and Tables

**Figure 1 cancers-11-00327-f001:**
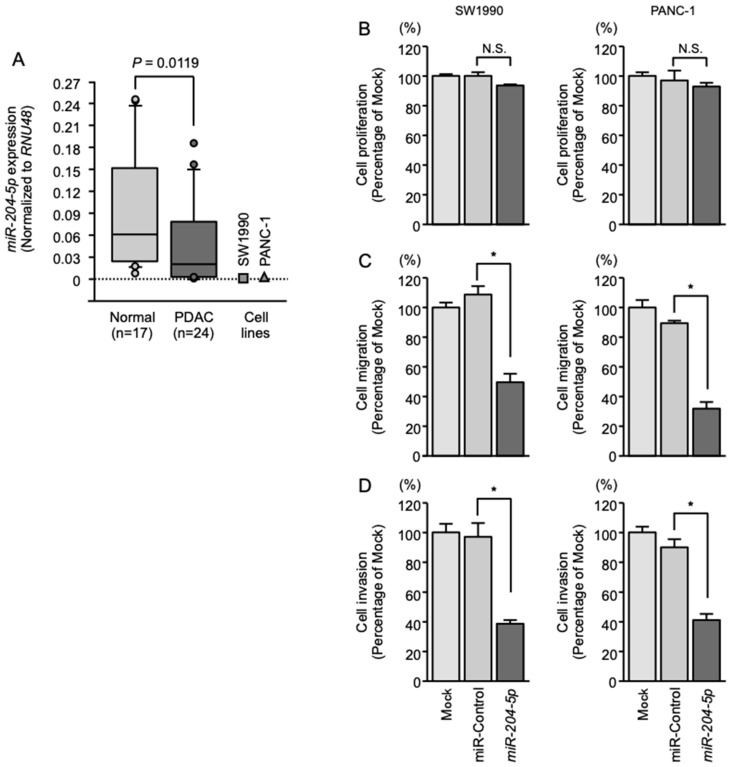
Downregulated expression and antitumor effect of *miR-204-5p* in pancreatic ductal adenocarcinoma (PDAC). (**A**) Expression levels of *miR-204-5p* in PDAC clinical specimens and cell lines determined by qRT-PCR. Expression levels were normalized to that of *RNU48*. *p* = 0.0119. (**B**) Cell proliferation determined by XTT assays 72 h after transfection of 10 nM *miR-204-5p*. * *p* < 0.0001; N.S.: not significant. (**C**) Cell migration assessed by migration assays. * *p* < 0.0001. (**D**) Cell invasion assessed by Matrigel invasion assays. * *p* < 0.0001.

**Figure 2 cancers-11-00327-f002:**
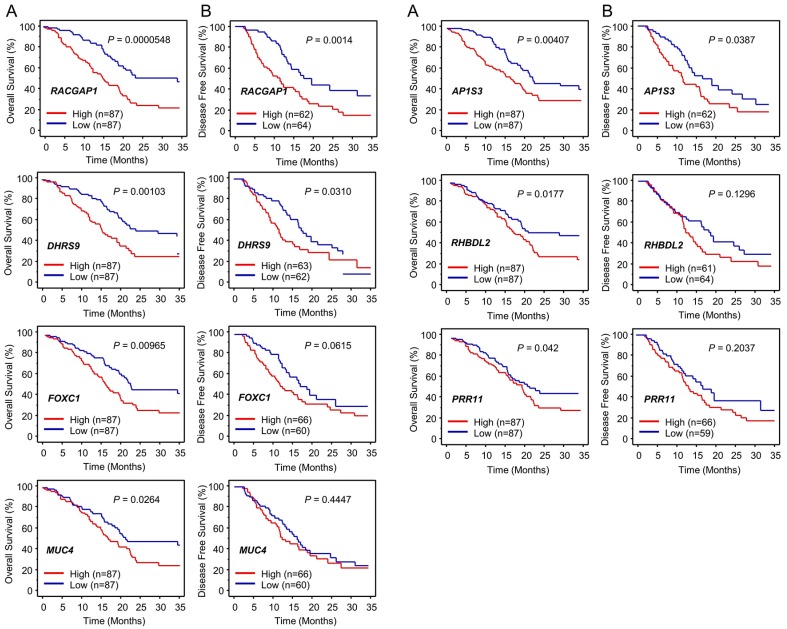
Clinical significance of the expression levels of seven genes targeted by *miR-204-5p* (*RACGAP1*, *DHRS9*, *AP1S3*, *FOXC1*, *PRP11*, *RHBDL2* and *MUC4*) based on data from The Cancer Genome Atlas (TCGA) database. (**A**) Kaplan-Meier plots of overall survival and (**B**) disease-free survival with log-rank tests comparing the survival of PDAC patients with high (red lines) versus low (blue lines) expression levels of each of the abovementioned genes, based on data from TCGA database.

**Figure 3 cancers-11-00327-f003:**
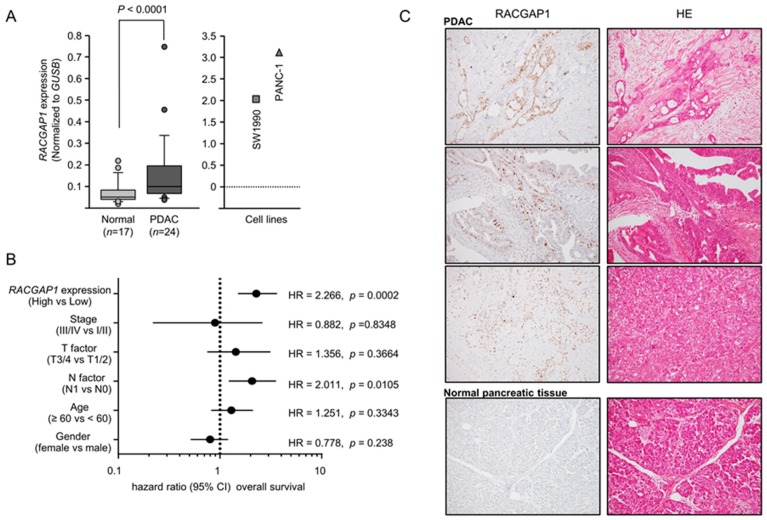
Aberrant expression of *RACGAP1* in PDAC specimens and its clinical significance. (**A**) Expression levels of *RACGAP1* in PDAC clinical specimens and two cell lines, SW1990 and PANC-1. *GUSB* expression was evaluated as the internal control. (**B**) Expression level of *RACGAP1* in patients with PDAC according to data from TCGA database (https://cancergenome.nih.gov/). Forest plot of univariate Cox proportional hazards regression analyses of 5-year overall survival rates according to *RACGAP1* expression and clinicopathological factors. (**C**) Immunohistochemical analysis of RACGAP1 in PDAC clinical samples. RACGAP1 was strongly expressed in cancer lesions. In normal pancreatic tissue, expression of RACGAP1 was not recognized. Original magnification: 200×.

**Figure 4 cancers-11-00327-f004:**
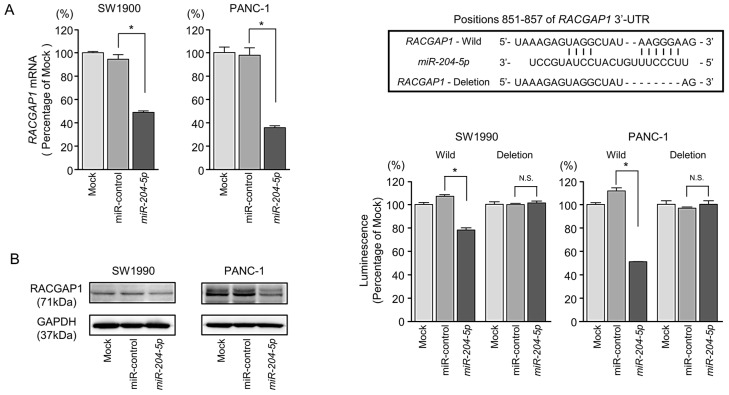
Direct regulation of *RACGAP1* by *miR-204-5p* in PDAC cells. (**A**) *RACGAP1* mRNA expression in PDAC cell lines was evaluated by qRT-PCR at 72 h after transfection of *miR-204-5p*. *GUSB* expression was evaluated as the internal control. * *p* < 0.0001. (**B**) RACGAP1 protein expression in PDAC cell lines was evaluated by western blotting at 96 h after transfection of *miR-204-5p*. GAPDH expression was evaluated as the loading control. (**C**) *miR-204-5p* binding sites within the 3′-UTR of *RACGAP1* mRNA. Dual luciferase reporter assays using vectors harboring either the wild-type *RACGAP1* 3′-UTR sequence containing the putative *miR-204-5p* binding site (nucleotide positions 851-857) or the 3′-UTR sequence with deletion of this binding site. Data represent the ratio of *Renilla*/*firefly* luciferase activities in PDAC cells transfected with *miR-204-5p* relative to that in mock-transfected control cells. * *p* < 0.0001; N.S.: no significant.

**Figure 5 cancers-11-00327-f005:**
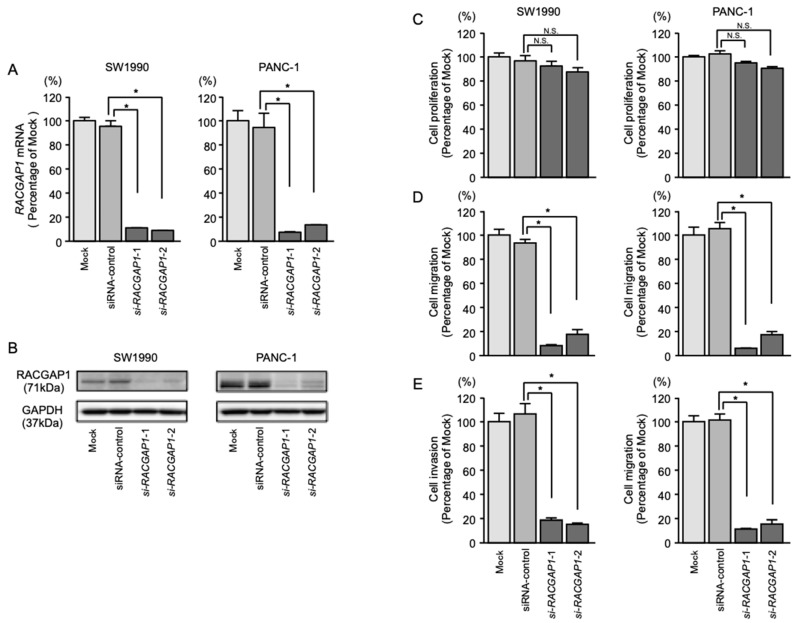
Effects of *RACGAP1* knockdown by si-*RACGAP1* transfection into PDAC cells. (**A**) *RACGAP1* mRNA expression in PDAC cell lines was evaluated by qRT-PCR at 72 h after transfection with si-*RACGAP1*-1 or si-*RACGAP1*-2. *GUSB* expression was evaluated as the internal control. (**B**) RACGAP1 protein expression in PDAC cell lines was evaluated by western blotting at 96 h after transfection with si-*RACGAP1-1* and si-*RACGAP1-2*. GAPDH expression was used as the loading control. (**C**) Cell proliferation was assessed by XTT assays 72 h after transfection with 10 nM si-*RACGAP1*-1 or si-*RACGAP1*-2. (**D**) Cell migration was assessed by migration assays. * *p* < 0.0001. (**E**) Cell invasion was assessed by Matrigel invasion assays. * *p* < 0.0001.

**Figure 6 cancers-11-00327-f006:**
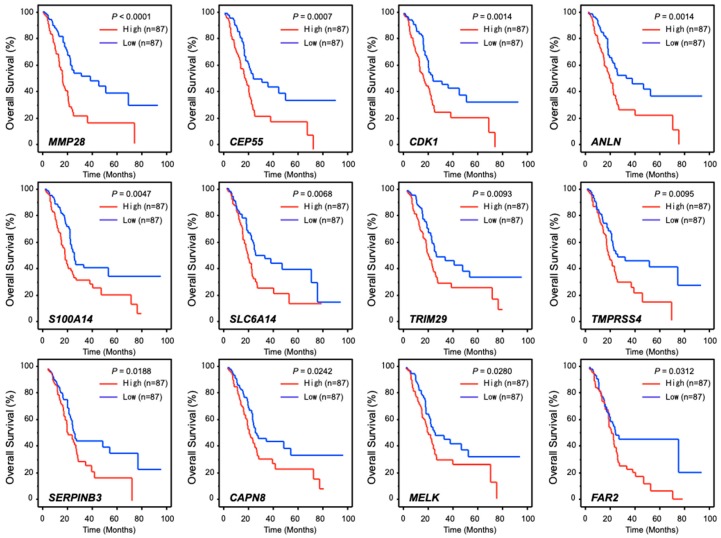
Clinical significance of *RACGAP1* downstream genes associated with a poor prognosis of patients with PDAC.

**Figure 7 cancers-11-00327-f007:**
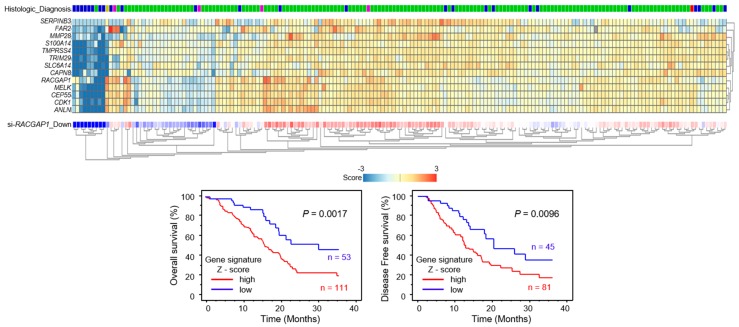
Combination analysis with heatmap of 12 target genes related to poor prognosis in PDAC. Heatmap was created using analysis webcite “R2: Genomics Analysis and Visualization Platform (http://r2.amc.nl)”. Z-score was evaluated by a combination of si-*RACGAP1* downstream genes based on TCGA datasets. High group (mRNA Z-score > 0) and low group (mRNA Z-score ≤ 0) are displayed as Kaplan–Meier plots with log-rank tests.

**Table 1 cancers-11-00327-t001:** Patient characteristics.

Pancreatic Ductal Adenocarcinoma (PDAC)	Factors	Number	(%)
Total number		24	
Average age(range), years		67.0 (42–79)	
Gender	Male	12	(50.0)
	Female	12	(50.0)
T category	pT1	1	(4.2)
	pT2	0	(0)
	pT3	23	(95.8)
	pT4	0	(0)
N category	0	8	(33.3)
	1	16	(66.7)
M category	0	23	(95.8)
	1	1	(4.2)
Lymphatic invasion	0	1	(4.2)
	1	11	(45.8)
	2	9	(37.5)
	3	3	(12.5)
Vascular invasion	0	1	(4.2)
	1	10	(41.7)
	2	11	(45.8)
	3	2	(8.3)
Neoadjuvant Chemotherapy	(−)	12	(50.0)
	(+)	12	(50.0)
Adjuvant Chemotherapy	(−)	7	(29.2)
	(+)	17	(70.8)
Recurrence	(−)	7	(29.2)
	(+)	17	(70.8)

**Table 2 cancers-11-00327-t002:** Patient characteristics.

Normal Pancreatic Tissue	Factors	Number	(%)
Total number		17	
Average age (range), years		64.8 (42–85)	
Gender	Male	6	(35.3)
	Female	11	(64.7)

**Table 3 cancers-11-00327-t003:** Candidate target genes regulated by *miR-204-5p*.

Entrez Gene ID	Gene Symbol	Gene Name	Target Sites		GEO	Array (SW1990)	OncoLnc OS_*p*-Value PDAC
			Conserved Sites	Poorly Sites	FC (log)	Mock vs. *miR-204-5p*	
29127	*RACGAP1*	Rac GTPase activating protein 1	0	1	1.34	−1.016	0.00005
10170	*DHRS9*	dehydrogenase/reductase (SDR family) member 9	0	2	1.57	−1.078	0.00103
130340	*AP1S3*	adaptor-related protein complex 1, sigma 3 subunit	0	2	1.17	−3.004	0.00407
2296	*FOXC1*	forkhead box C1	1	1	1.01	−1.059	0.00965
54933	*RHBDL2*	rhomboid, veinlet-like 2 (Drosophila)	0	1	1.18	−1.061	0.01770
4585	*MUC4*	mucin 4, cell surface associated	0	1	1.42	−1.569	0.02640
55771	*PRR11*	proline rich 11	1	0	1.42	−1.406	0.04200
4680	*CEACAM6*	carcinoembryonic antigen-related cell adhesion molecule 6 (non-specific cross reacting antigen)	0	1	3.50	−1.260	0.07160
6505	*SLC1A1*	solute carrier family 1 (neuronal/epithelial high affinity glutamate transporter, system Xag), member 1	0	2	1.52	−1.397	0.08320
2335	*FN1*	fibronectin 1	0	1	3.73	−1.023	0.09500
3397	*ID1*	inhibitor of DNA binding 1, dominant negative helix-loop-helix protein	0	1	1.29	−1.384	0.15300
55808	*ST6GALNAC1*	ST6 (α-N-acetyl-neuraminyl-2,3-β-galactosyl-1,3)-N-acetylgalactosaminide alpha-2,6-sialyltransferase 1	0	1	1.36	−1.367	0.15300
54210	*TREM1*	triggering receptor expressed on myeloid cells 1	0	1	1.55	−1.067	0.15700
8905	*AP1S2*	adaptor-related protein complex 1, sigma 2 subunit	2	0	1.04	−1.806	0.17700
147495	*APCDD1*	adenomatosis polyposis coli down-regulated 1	0	1	1.18	−1.276	0.21200
4837	*NNMT*	nicotinamide N-methyltransferase	0	1	2.69	−1.060	0.21900
7851	*MALL*	mal, T-cell differentiation protein-like	1	1	1.44	−1.252	0.22600
493	*ATP2B4*	ATPase, Ca++ transporting, plasma membrane 4	0	1	1.35	−1.149	0.23600
1295	*COL8A1*	collagen, type VIII, α 1	0	1	4.62	−1.131	0.36400
219699	*UNC5B*	unc-5 homolog B (C. elegans)	1	1	1.42	−1.628	0.59600
2192	*FBLN1*	fibulin 1	0	1	1.62	−1.262	0.65300
5159	*PDGFRB*	Platelet-derived growth factor receptor, beta polypeptide	0	2	1.80	−2.235	0.77500
2182	*ACSL4*	acyl-CoA synthetase long-chain family member 4	1	2	1.05	−1.570	0.79200
140885	*SIRPA*	signal-regulatory protein alpha	0	1	1.20	−1.194	0.80000
1809	*DPYSL3*	dihydropyrimidinase-like 3	1	1	2.07	−1.185	0.99600

GEO: Gene Expression Omnibus; FC: Fold-Change; OS: Overall Survival; PDAC: Pancreatic Ductal Adenocarcinoma.

**Table 4 cancers-11-00327-t004:** *RACGAP1*-meditade downstream genes in PDAC.

Entrez GeneID	Gene Symbol	GeneName	*miR-204-5p* Conserved Sites Total	*miR-204-5p* Poorly Conserved Sites Total	GEO FC (log)	Array(SW1990) Mock vs. si-*RACGAP1*	OncoLnc OS_*p*-Value PAAD
79148	*MMP28*	matrix metallopeptidase 28	-	-	1.44652	−1.30890	1.45E-05
29127	*RACGAP1*	Rac GTPase activating protein 1	0	1	1.34429	−3.04118	0.0000548
55165	*CEP55*	centrosomal protein 55 kDa	-	-	1.09192	−1.05821	0.000736
983	*CDK1*	cyclin-dependent kinase 1	-	-	1.41425	−1.03595	0.00137
54443	*ANLN*	anillin, actin binding protein	-	-	1.72921	−1.19268	0.00142
57402	*S100A14*	S100 calcium binding protein A14	-	-	1.06798	−1.19808	0.00469
11254	*SLC6A14*	solute carrier family 6 (amino acid transporter), member 14	0	1	3.01708	−1.20549	0.00678
23650	*TRIM29*	tripartite motif containing 29	-	-	2.00490	−1.23048	0.00934
56649	*TMPRSS4*	transmembrane protease, serine 4	-	-	1.96115	−1.27289	0.00954
6317	*SERPINB3*	serpin peptidase inhibitor, clade B (ovalbumin), member 3	-	-	1.74282	−1.15475	0.0188
388743	*CAPN8*	calpain 8	-	-	1.92729	−2.40223	0.0242
9833	*MELK*	maternal embryonic leucine zipper kinase	0	1	1.21153	−1.04052	0.028
55711	*FAR2*	fatty acyl CoA reductase 2	-	-	1.04894	−1.13250	0.0312
1893	*ECM1*	extracellular matrix protein 1	-	-	1.14510	−1.03234	0.0531
11178	*LZTS1*	leucine zipper, putative tumor suppressor 1	-	-	1.30061	−1.64372	0.0591
6696	*SPP1*	secreted phosphoprotein 1	-	-	2.67411	−1.79208	0.0639
5349	*FXYD3*	FXYD domain containing ion transport regulator 3	-	-	1.80037	−1.46428	0.0677
4680	*CEACAM6*	carcinoembryonic antigen-related cell adhesion molecule 6 (non-specific cross reacting antigen)	0	1	3.49703	−1.04848	0.0716
80736	*SLC44A4*	solute carrier family 44, member 4	-	-	1.84270	−1.14495	0.0761
80856	*KIAA1715*	KIAA1715	0	2	1.05264	−1.70461	0.0891
412	*STS*	steroid sulfatase (microsomal), isozyme S	-	-	1.32435	−2.62844	0.0996
1087	*CEACAM7*	carcinoembryonic antigen-related cell adhesion molecule 7	-	-	1.89107	−1.11293	0.105
29969	*MDFIC*	MyoD family inhibitor domain containing	0	1	1.20813	−1.20299	0.11
23596	*OPN3*	opsin 3	-	-	1.28539	−1.53133	0.122
56241	*SUSD2*	sushi domain containing 2	-	-	1.30384	−1.26476	0.123
55808	*ST6GALNAC1*	ST6 (alpha-N-acetyl-neuraminyl-2,3-β-galactosyl-1,3)-N-acetylgalactosaminide α-2,6-sialyltransferase 1	0	1	1.35766	−1.55278	0.153
90459	*ERI1*	exoribonuclease 1	-	-	1.08766	−1.07601	0.158
1999	*ELF3*	E74-like factor 3 (ETS domain transcription factor, epithelial-specific)	-	-	1.18538	−1.05779	0.174
121457	*IKBIP*	IKBKB interacting protein	-	-	1.10964	−1.06344	0.185
50810	*HDGFRP3*	hepatoma-derived growth factor, related protein 3	-	-	1.37815	−1.69105	0.187
1048	*CEACAM5*	carcinoembryonic antigen-related cell adhesion molecule 5	-	-	2.76693	−1.96132	0.188
147495	*APCDD1*	adenomatosis polyposis coli down-regulated 1	0	1	1.17965	−1.33923	0.212
200958	*MUC20*	mucin 20, cell surface associated	-	-	1.50494	−1.28329	0.232
54843	*SYTL2*	synaptotagmin-like 2	-	-	1.90048	−1.17307	0.242
7031	*TFF1*	trefoil factor 1	-	-	2.40095	−1.13688	0.242
1847	*DUSP5*	dual specificity phosphatase 5	-	-	1.47556	−1.77597	0.269
85477	*SCIN*	scinderin	-	-	1.14779	−1.62677	0.285
10000	*AKT3*	v-akt murine thymoma viral oncogene homolog 3	-	-	1.02342	−1.07780	0.317
89932	*PAPLN*	papilin, proteoglycan-like sulfated glycoprotein	0	2	1.81417	−1.08233	0.332
51056	*LAP3*	leucine aminopeptidase 3	-	-	1.06387	−1.23137	0.343
4688	*NCF2*	neutrophil cytosolic factor 2	-	-	2.00063	−1.14183	0.422
22795	*NID2*	nidogen 2 (osteonidogen)	-	-	1.78329	−1.09923	0.425
8706	*B3GALNT1*	β-1,3-N-acetyl-galactosaminyl-transferase 1 (globoside blood group)	-	-	1.06403	−1.07457	0.474
10551	*AGR2*	anterior gradient 2	-	-	2.04850	−1.20759	0.491
1009	*CDH11*	cadherin 11, type 2, OB-cadherin (osteoblast)	1	0	3.64065	−1.32668	0.526
340547	*VSIG1*	V-set and immunoglobulin domain containing 1	-	-	1.78237	−1.44981	0.54
219699	*UNC5B*	unc-5 homolog B (*C. elegans*)	1	1	1.42242	−1.48944	0.596
154141	*MBOAT1*	membrane bound O-acyltransferase domain containing 1	-	-	1.03451	−1.03993	0.615
3455	*IFNAR2*	interferon (α, β and ω) receptor 2	0	3	1.07229	−1.11007	0.618
2357	*FPR1*	formyl peptide receptor 1	0	2	1.09684	−1.92665	0.629
29887	*SNX10*	sorting nexin 10	-	-	1.20341	−1.63818	0.64
91607	*SLFN11*	Schlafen family member 11	-	-	1.37401	−1.23024	0.661
79071	*ELOVL6*	ELOVL fatty acid elongase 6	1	2	1.48750	−1.18773	0.727
3075	*CFH*	complement factor H	-	-	1.64335	−1.25960	0.732
5159	*PDGFRB*	platelet-derived growth factor receptor, β polypeptide	0	2	1.79906	−3.93598	0.775
10855	*HPSE*	heparanase	-	-	1.10037	−1.25724	0.783
2877	*GPX2*	glutathione peroxidase 2 (gastrointestinal)	-	-	2.06843	−1.24329	0.84
634	*CEACAM1*	carcinoembryonic antigen-related cell adhesion molecule 1 (biliary glycoprotein)	-	-	1.41681	−1.08376	0.843
54749	*EPDR1*	ependymin related 1	-	-	1.00831	−1.25356	0.86
2124	*EVI2B*	ecotropic viral integration site 2B	-	-	1.54469	−1.32368	0.939
3310	*HSPA6*	heat shock 70kDa protein 6 (HSP70B’)	-	-	1.34685	−1.10270	0.956
6453	*ITSN1*	intersectin 1 (SH3 domain protein)	-	-	1.21741	−1.04096	0.982
30001	*ERO1L*	ERO1-like (S. cerevisiae)	0	1	1.05476	−1.33509	No data
100499177	*THAP9-AS1*	THAP9 antisense RNA 1	-	-	1.02990	−1.06954	No data

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
