# Peer review of "Gene Regulation by Antitumor miR-204-5p in Pancreatic Ductal Adenocarcinoma: The Clinical Significance of Direct RACGAP1 Regulation"

_cancers, 2019, doi:10.3390/cancers11030327_

Reviewer 1 Report

It is an interesting study however the authors support their study without using recent important literature regarding GWAS on PDAC and they did not underlie the differences between Asian and European ancestry populations on genetic risk for PDAC. Related references like   Pancreas. 2013 Jan;42(1):67-71; Int J Cancer. 2015 Nov 1;137(9):2175-83Nat Genet. 2015 Aug;47(8):911-6Oncotarget. 2016 Oct 11;7(41):66328-66343Nat Commun. 2018 Feb 8;9(1):556. should be included and discussed

Author Response

Cancers Manuscript ID: cancers-437096

Dear Dr. Osaki

On behalf of my co-authors, I would like to revise an original research article entitled, “Gene regulation by antitumor miR-204-5p in pancreatic ductal adenocarcinoma: direct regulation of RACGAP1 and its clinical significance” for consideration for publication in Cancers, Special Issue: MicroRNA-Associated Cancer Metastasis.

Reviewer #1

Comments and Suggestions for Authors:

It is an interesting study however the authors support their study without using recent important literature regarding GWAS on PDAC and they did not underlie the differences between Asian and European ancestry populations on genetic risk for PDAC. Related references like Pancreas. 2013 Jan;42(1):67-71; 

Int J Cancer. 2015 Nov 1;137(9):2175-83, 

Nat Genet. 2015 Aug;47(8):911-6, 

Oncotarget. 2016 Oct 11;7(41):66328-66343, 

Nat Commun. 2018 Feb 8;9(1):556.

should be included and discussed.

Response:

As suggested the reviewer’s comment, I added following sentences in “Introduction” and cited articles were added.

Developing high-throughput genotyping technologies have been accelerated personalized medicine. Through the latest Genome-Wide Association Studies (GWAS) assays, several genetic risk variants involved in the risk of PDAC developing have been detected [6-9]. Importantly, GWAS catalogue shows that functional genetic variants revealed by GWAS were localized in non-coding region of human genome. In the near future, GWAS analyses may show variants of miRNAs and risk of developing PDAC.

Our data are the first to report these findings in PDAC.

We believe that this article contributes significantly to our understanding of the molecular mechanisms of PDAC pathogenesis. We appreciate your consideration of this manuscript for publication in Cancers, Special Issue: MicroRNA-Associated Cancer Metastasis.

Sincerely,

Naohiko Seki, PhD

Department of Functional Genomics

Graduate School of Medicine, Chiba University

1-8-1 Inohana, Chuo-ku, Chiba 260-8670, Japan

Reviewer 2 Report

In the manuscript entitled "Gene regulation by antitumor miR-204-5p in pancreatic ductal adenocarcinoma: the clinical significance of direct RACGAP1 regulation", Khalid et al found that the expression of miR-204-5p in PDAC clinical samples was down regulated. Using high-throughput analysis, they were able to identify several putative targets of this miRNA. The authors specifically validated RACGAP1 as a target of miR-204-5p, and found that high expression of RACGAP1 is negatively associated with PDAC patient survival. Overall, this study provided solid evidence on miR-204-5p's role in PDAC carcinogenesis, and suggested miR-204-5p and its target RACGAP1 as potential therapeutic targets for PDAC treatment. The quality of this study meets the standards of Cancers, and it can be considered for publication after minor revisions. Below are several suggestions I have for this manuscript:

In Figure 3C, the authors showed IHC staining of RACGAP1 in cancer lesions. They should also show the IHC results for normal tissues if it is available.

Since RACGAP1 is the top hit of miR-204-5p targets, the authors can explore a bit more on this gene: is there evidence to show any gain-of-function mutation or chromosomal ploidy alterations of RACGAP1 in PDAC patients? What about other cancers? Also, for the RACGAP1 downstream genes reported in Table 3, has any of these regulations been reported in other literatures?

The Table 2 and Table 3 can only include statistically significant hits (p value less than 0.05). The full list can be attached in supplementary materials.

Author Response

Cancers Manuscript ID: cancers-437096

Dr. Mitsuhiko Osaki

Guest Editor

Dr. Serene Tian

Assistant Editor

Cancers: Special Issue "MicroRNA-Associated Cancer Metastasis"

Dear Dr. Osaki

On behalf of my co-authors, I would like to revise an original research article entitled, ““Gene regulation by antitumor miR-204-5p in pancreatic ductal adenocarcinoma: direct regulation of RACGAP1 and its clinical significance” for consideration for publication in Cancers, Special Issue: MicroRNA-Associated Cancer Metastasis.

Reviewer #2

Comments and Suggestions for Authors

In the manuscript entitled "Gene regulation by antitumor miR-204-5p in pancreatic ductal adenocarcinoma: the clinical significance of direct RACGAP1 regulation", Khalid et al found that the expression of miR-204-5p in PDAC clinical samples was down regulated. Using high-throughput analysis, they were able to identify several putative targets of this miRNA. The authors specifically validated RACGAP1 as a target of miR-204-5p, and found that high expression of RACGAP1 is negatively associated with PDAC patient survival. Overall, this study provided solid evidence on miR-204-5p's role in PDAC carcinogenesis, and suggested miR-204-5p and its target RACGAP1 as potential therapeutic targets for PDAC treatment. The quality of this study meets the standards of Cancers, and it can be considered for publication after minor revisions. Below are several suggestions I have for this manuscript:

Comment-1:

In Figure 3C, the authors showed IHC staining of RACGAP1 in cancer lesions. They should also show the IHC results for normal tissues if it is available.

Response:

As suggested by the reviewer’s comment, I added IHC image of normal tissue and HE staining was added to the Figure 3.

Comment-2:

Since RACGAP1 is the top hit of miR-204-5p targets, the authors can explore a bit more on this gene: is there evidence to show any gain-of-function mutation or chromosomal ploidy alterations of RACGAP1 in PDAC patients? What about other cancers? Also, for the RACGAP1 downstream genes reported in Table 3, has any of these regulations been reported in other literatures?

Response:

According to the reviewer’s comment, I searched previous studies on RACGAP1 mutation. For PDAC, genomic alteration in the chromosomal region of RACGAP1 (12q13 region) was not reported.

The clinical significance of RACGAP1 expression in human cancers was investigated by using TCGA database. High expression of RACGAP1 was associated with poor prognosis of several cancers, e.g., kidney renal papillary cell carcinoma, hepatocellular carcinoma, lung adenocarcinoma, sarcoma and skin cutaneous melanoma. New data was added in Supplemental Figure 5 and mention as follows in Results 2.4.

In addition, TCGA database analyses showed that high expression of RACGAP1 was closely associated with poor prognosis (3 year overall survival) of several cancers, e.g., kidney renal papillary cell carcinoma, hepatocellular carcinoma, lung adenocarcinoma, sarcoma and skin cutaneous melanoma (Supplemental Figure 5).

As suggested by the reviewer’s comment, I added following sentences in Discussion last chapter”.

Interestingly, TMPRSS4 and SERPINB3 were identified as a PDAC classifier for discriminating PDAC and early precursor lesions from non-malignant tissue [40]. Notably, silencing of TMPRESS4 expression blocked PDAC cell migration and invasion abilities [40]. Among these targets, we searched for regulation of miRNAs. Aberrant expression of the actin-binding protein ANLN (anillin) was detected in PDAC clinical specimens, and knockdown of ANLN markedly inhibited the migration and invasion of PDAC cell lines [17]. In addition, that same study found that ANLN was directly controlled by antitumor miR-217 in PDAC cells. Ectopic expression of miR-217 was significantly blocked cancer cell aggressiveness features [17]. Moreover, TRIM29, DNA and p53-binding protein is overexpressed in PDAC and its function as a mediator in DNA damage signaling [41]. Aberrant expression of TRIM29 associated with radio-resistance in PDAC cells [41]. Tumor suppressive miR-449a inhibited PDAC cell aggressiveness through targeting TRIM29 [41]. These results suggest that RACGAP1 and RACGAP1-mediated genes are possible therapeutic targets for PDAC. Analysis strategy based on aberrant expressed miRNAs is effective for searching to novel molecular pathogenesis of PDAC.

Comment-3:

The Table 2 and Table 3 can only include statistically significant hits (p value less than 0.05). The full list can be attached in supplementary materials.

Response:

Reviewer's indication is important point to researchers. However, the description of all genes is not efficient for subsequent analysis. Our genome-wide gene expression analyses data (miR-204-5p targets and RACGAP1 mediated genes) was registered in the GEO database (accession numbers, GSE115801 and GSE115909). Researchers can access and analyze our data at any time. The accession numbers are listed in the text.

Our data are the first to report these findings in PDAC.

We believe that this article contributes significantly to our understanding of the molecular mechanisms of PDAC pathogenesis. We appreciate your consideration of this manuscript for publication in Cancers, Special Issue: MicroRNA-Associated Cancer Metastasis.

Sincerely,

Naohiko Seki, PhD

Department of Functional Genomics

Graduate School of Medicine, Chiba University

1-8-1 Inohana, Chuo-ku, Chiba 260-8670, Japan

Reviewer 3 Report

Khalid Muhammad et al have elaborated the significance of their previous observation that miR-204-5p down regulation is associated with bad prognosis in PDAC. The authors observed that miR-204-5p, which is tumorigenic in other cancers had anti-tumor properties in PDAC. They identified target genes of this miRNA, specifically highlighting the significance of RACGAP1, which is over expressed in PDAC. The experiments performed are adequate to identify the role of RACGAP1 as a possible target for anti-PDAC therapy. 

Author Response

Cancers Manuscript ID: cancers-437096

Dr. Mitsuhiko Osaki

Guest Editor

Dr. Serene Tian

Assistant Editor

Cancers: Special Issue "MicroRNA-Associated Cancer Metastasis"

Dear Dr. Osaki

On behalf of my co-authors, I would like to revise an original research article entitled, “Gene regulation by antitumor miR-204-5p in pancreatic ductal adenocarcinoma: direct regulation of RACGAP1 and its clinical significance” for consideration for publication in Cancers, Special Issue: MicroRNA-Associated Cancer Metastasis.

Reviewer #3

Comments and Suggestions for Authors:

Khalid Muhammad et al have elaborated the significance of their previous observation that miR-204-5p down regulation is associated with bad prognosis in PDAC. The authors observed that miR-204-5p, which is tumorigenic in other cancers had anti-tumor properties in PDAC. They identified target genes of this miRNA, specifically highlighting the significance of RACGAP1, which is over expressed in PDAC. The experiments performed are adequate to identify the role of RACGAP1 as a possible target for anti-PDAC therapy.

Response:

We appreciate your review.

We believe that this study may have innovative effects on pancreatic cancer molecular science. As other reviewers point out, there are additional correction points, so I'd appreciate its if you can confirm.

Our data are the first to report these findings in PDAC.

We believe that this article contributes significantly to our understanding of the molecular mechanisms of PDAC pathogenesis. We appreciate your consideration of this manuscript for publication in Cancers, Special Issue: MicroRNA-Associated Cancer Metastasis.

Sincerely,

Naohiko Seki, PhD

Department of Functional Genomics

Graduate School of Medicine, Chiba University

1-8-1 Inohana, Chuo-ku, Chiba 260-8670, Japan

Reviewer 4 Report

In the present manuscript, entitled “Gene regulation by antitumor miR-204-5p
in pancreatic ductal adenocarcinoma: the clinical significance of direct RACGAP1
regulation” Muhammad Khalid and co authors evaluating miR-204-5p expression in
pancreatic ductal adenocarcinoma specimens found a significant down regulation
both in tissues than in cell lines comparing miR expression to normal pancreatic
tissues. Ectopic expression of miR-204-5p affected cell migration and invasion of
SW1990 and PANC-1 cell lines but doesn’t changed cell proliferation.

To identified putative oncogenes regulated by miR-204-5p in pancreatic ductal
adenocarcinoma, the authors combined data obtained by comprehensive gene
expression analysis of SW1990 transfected with miR-204-5p with an in silico
analysis performed using TargetScanHuman 7.2. A list of 25 deregulated genes
was obtained matching this data with their clinical relevance, expression and
prognosis in pancreatic ductal adenocarcinoma patients from TCGA database.
Since RACGAP1 arise as significantly associated with poor prognosis of patients
affected by pancreatic ductal adenocarcinoma, Muhammad Khalid and co authors,
evaluated by univariate and multivariate Cox hazard regression analysis the clinical
relevance of RACGAP1 using TCGA database patients, and assessed by
immunohistochemical analysis the protein level in pancreatic ductal
adenocarcinoma and noncancerous epithelial tissues.

Overexpression of miR-204-5p reduces mRNA and protein expression of
RACGAP1 and Luciferase assay proved the direct miR-204-5p binding on 3’UTR of
RACGAP1. RACGAP1 siRNA transfection elicited a perturbation of cell migration
and invasion but didn’t affect cell proliferation. As well as performed to discovery
miR-204-5p target genes, the authors choose to identify RACGAP1 downstream
targets genes interfering protein expression in SW1990 cell lines and analysing
genome-wide gene expression. 64 putative high expressed target genes were then
associated with overall survival of pancreatic ductal adenocarcinoma patients from
TCGA database.

In the opinion of this reviewer to be considered for publication the following
manuscript needs to be improved in the following point:

Since Muhammad Khalid et al., demonstrated that ectopic expression of miR-204-
5p as well as RACGAP1 inhibition affect cell migration and invasion of SW1990 and
PANC1 cell lines, should the authors perform a pathways prediction analysis using
the candidate miR-204-5p targets genes identified?

Should the authors assess the modulation of known migration and invasion markers
such as vimentin, E-cadherin, fibronectin, N-cadherin, in SW1990 and PANC1 cell
lines transfected with miR-204-5p and interfered for RACGAP1?

Should the authors specify how many samples did they analyse for RACGAP1
staining by immunohistochemical assay’? Should the authors add a graph showing
the positive cell count? To corroborate the anti-correlation between RACGAP1 and
miR-204-5p expression, should the authors assess the level of the miR-204-5p in
the same tissues? Should the authors correlate this data with any clinic-pathological
factors?

To validate the data obtained by both the genome wide gene expression analysis
performed, should the authors add qRT-PCR of some targets identified.
In the latest part of the manuscript the authors identified 64 putative genes affected
by RACGAP1 in SW1990 but they didn’t provide any experiments to prove this
results.

Should the authors analyse the expression of some of these targets in the tissues
samples and in TCGA database?

Since the authors demonstrate that RACGAP1 interference impinges on migration
and invasion of pancreatic ductal adenocarcinoma cell, should Muhammad Khalid
et al., demonstrate that overexpression of almost one of targets genes identified
was able to revert this phenotype?

Should the authors assess if these targets have a predicted binding region for miR-
204-5p?

Author Response

Cancers Manuscript ID: cancers-437096

Dr. Mitsuhiko Osaki

Guest Editor

Dr. Serene Tian

Assistant Editor

Cancers: Special Issue "MicroRNA-Associated Cancer Metastasis"

Dear Dr. Osaki

On behalf of my co-authors, I would like to revise an original research article entitled, ““Gene regulation by antitumor miR-204-5p in pancreatic ductal adenocarcinoma: direct regulation of RACGAP1 and its clinical significance” for consideration for publication in Cancers, Special Issue: MicroRNA-Associated Cancer Metastasis.

Comment-1:

In the opinion of this reviewer to be considered for publication the following manuscript needs to be improved in the following point: Since Muhammad Khalid et al., demonstrated that ectopic expression of miR-204-5p as well as RACGAP1 inhibition affect cell migration and invasion of SW1990 and PANC1 cell lines, should the authors perform a pathways prediction analysis using the candidate miR-204-5p targets genes identified?

Response:

As suggested by the reviewer’s comment, I performed pathway analyses using miR-204-5p target genes and RACGAP1 mediated downstream genes. The newly analyzed data were shown in the Supplemental Tables 1 and 2. And I mentioned as follows in RESULTS 2.3 and 2.7.

We also investigated the pathway analyses using miR-204-5p controlled downregulated genes. Several pathways were identified as miR-204-5p controlled pathways, e.g., “Regulation of actin cytoskeleton”, “Cytokine-cytokine receptor interaction”, “MAPK signaling pathway” and “Endocytosis” (Supplemental Table 1).

To explore the molecular network controlled by RACGAP1 in PDAC cells, we performed pathway analyses using RACGAP1 mediated downregulated genes (Supplemental Table 2).

Comment-2:

Should the authors assess the modulation of known migration and invasion markers such as vimentin, E-cadherin, fibronectin, N-cadherin, in SW1990 and PANC1 cell lines transfected with miR-204-5p and interfered for RACGAP1?

Response:

As suggested by the reviewer’s comment, I picked up the EMT-related genes based on array data (gene expression data of ectopic expression of miR-204-5p or transfection of si-RACGAP1 in SW1990 cells). The newly analyzed data were shown in the Supplemental Tables 3 and 4. And I mentioned as follows in RESULTS 2.6.

EMT-related genes were selected and picked up from gene expression analysis data (GSE115801 and GSE115909). Our array data showed that expression of SNAL3 and vimentin were reduced by si-RACGAP1 transfection into SW1990 cells. Expression of fibronectin 1 was reduced by miR-204-5p transfection into SW1990 cells (Supplemental Tables 3 and 4).

Comment-3:

Should the authors specify how many samples did they analyse for RACGAP1 staining by immunohistochemical assay’? Should the authors add a graph showing the positive cell count? To corroborate the anti-correlation between RACGAP1 and miR-204-5p expression, should the authors assess the level of the miR-204-5p in the same tissues? Should the authors correlate this data with any clinic-pathological factors?

Response:

I appreciate your informative comments. Then, I will state my stance on IHC of RACGAP1 in this paper. Is the RACGAP1 (miR-204-5p target gene) actually expressed in cancer cells? I confirmed that by IHC. Clinicopathological analysis by immunostaining is not done in this paper. I appreciate your understanding.

Negative correlation of miR-204-5p and RACGAP1 expression in PDAC clinical specimens was investigated by our 24 PDAC specimens. The newly analyzed data were shown in the Supplemental Figure 2, and I mentioned as follows in RESULTS 2.4.

Negative correlations between RACGAP1 mRNA expression and miR-204-5p expression were analyzed by Spearman’s rank test (R = -0.381, P < 0.0070, Supplemental Figure 2).

Also, clinical significance of expression of miR-204-5p and RACGAP1 in PDAC was investigated by using TCGA database. The newly analyzed data were shown in the Supplemental Figures 3 and 4, and I mentioned as follows in RESULTS 2.4.

We extracted samples of PDAC from the TCGA database. We analyzed clinicopathological factors of miR-204-5p and RACGAP1 expression (miR-204-5p; Supplemental Figures 3 and RACGAP1; Supplemental Figures 4). The recurrence of RACGAP1 showed high expression with a significant difference (P < 0.0015).

Comment-4:

To validate the data obtained by both the genome wide gene expression analysis performed, should the authors add qRT-PCR of some targets identified.

Response:

I appreciate your informative comments. The reviewer's point of view is currently underway of our research, then we want to avoid presenting data. The usefulness and credibility of Table 2 are showed recent papers. I added following sentences in Discussion 4th chapter.

Previous studies showed that FOXC1 was directly regulated by antitumor miR-204-5p in laryngeal squamous cell carcinoma and endometrial cancer [25]. Furthermore, AP1S3 was directly regulated by miR-204-5p in TNBC cells [24]

Comment-5:

In the latest part of the manuscript the authors identified 64 putative genes affected by RACGAP1 in SW1990 but they didn’t provide any experiments to prove this results. Should the authors analyse the expression of some of these targets in the tissues samples and in TCGA database?

Response:

As suggested by the reviewer’s comment, I investigated the clinical significance of PDAC using RACGAP1 mediated downstream genes. The newly analyzed data were shown in Figure 7, and I mentioned as follows in RESULTS 2.7 and MATERIALS AND METHODS 4.10.

Furthermore, we provided a heatmap gene visualization and validated as a prognostic ability of these 12 genes (Figure 7). As shown in Figure 7, patients with high gene signature expressions (Z-score > 0) were significantly poor OS and DFS rate than those with low gene signature expressions (Z-score ≤ 0) (OS; P = 0.0017, DFS; P = 0.0096, Figure 7). To explore the molecular network controlled by RACGAP1 in PDAC cells, we performed pathway analyses using RACGAP1 mediated downregulated genes (Supplemental Table 2).

The Z-scores of target genes mRNA expression data and clinical sample information corresponding to PDAC patients were collected from cBioPortal. R2: Genomics Analysis and Visualization Platform (http://r2.amc.nl) was used to create a heatmap. Furthermore, Z- score was evaluated by a combination of each genes sets. High group (mRNA Z-score > 0) and low group (mRNA Z-score ≤ 0) were analyzed by Kaplan–Meier survival curves and log-rank statistics.

Comment-6:

Since the authors demonstrate that RACGAP1 interference impinges on migration and invasion of pancreatic ductal adenocarcinoma cell, should Muhammad Khalidet al., demonstrate that overexpression of almost one of targets genes identified was able to revert this phenotype?

Response:

The analysis pointed out by the reviewer is an important issue in this study. Expression of RACGAP1 in pancreatic cancer cell lines was highly expressed. Therefore, transient expression of RACGAP1 could not find changes in advanced phenotypes in cell lines. We are studying overexpression of RACGAP1 enhancing malignant phenotypes and activation oncogenic signaling in several cancers including PDAC. Data cannot be presented in this paper. I appreciate your understanding.

Comment-7:

Should the authors assess if these targets have a predicted binding region for miR-204-5p?

Response:

As suggested by the reviewer’s comment, I added the prediction of miR-204-5p binding sites in their genes in Table 3.

Our data are the first to report these findings in PDAC.

We believe that this article contributes significantly to our understanding of the molecular mechanisms of PDAC pathogenesis. We appreciate your consideration of this manuscript for publication in Cancers, Special Issue: MicroRNA-Associated Cancer Metastasis.

Sincerely,

Naohiko Seki, PhD

Department of Functional Genomics

Graduate School of Medicine, Chiba University

1-8-1 Inohana, Chuo-ku, Chiba 260-8670, Japan

Round  2

Reviewer 1 Report

The authors cover all the revision issues

Author Response

Reviewer Report (Round 2)

Cancers Manuscript ID: cancers-437096

Dr. Mitsuhiko Osaki

Guest Editor

Dr. Serene Tian

Assistant Editor

Cancers: Special Issue "MicroRNA-Associated Cancer Metastasis"

Dear Dr. Osaki

On behalf of my co-authors, I would like to revise an original research article entitled, ““Gene regulation by antitumor miR-204-5p in pancreatic ductal adenocarcinoma: direct regulation of RACGAP1 and its clinical significance” for consideration for publication in Cancers, Special Issue: MicroRNA-Associated Cancer Metastasis.

I appreciate for your detailed review and comments regarding our paper. Your comments will be very useful for our paper.

Our data are the first to report these findings in PDAC.

We believe that this article contributes significantly to our understanding of the molecular mechanisms of PDAC pathogenesis. We appreciate your consideration of this manuscript for publication in Cancers, Special Issue: MicroRNA-Associated Cancer Metastasis.

Sincerely,

Naohiko Seki, PhD

Department of Functional Genomics

Graduate School of Medicine, Chiba University

1-8-1 Inohana, Chuo-ku, Chiba 260-8670, Japan

Reviewer 4 Report

in the opinion of this reviewer the authors have partially addressed the concerns raised by this reviewer in the first round of review adding in silico analysis but they didn't provide the new experiments required. 

Author Response

Reviewer Report (Round 2)

Cancers Manuscript ID: cancers-437096

Dr. Mitsuhiko Osaki

Guest Editor

Dr. Serene Tian

Assistant Editor

Cancers: Special Issue "MicroRNA-Associated Cancer Metastasis"

Dear Dr. Osaki

On behalf of my co-authors, I would like to revise an original research article entitled, ““Gene regulation by antitumor miR-204-5p in pancreatic ductal adenocarcinoma: direct regulation of RACGAP1 and its clinical significance” for consideration for publication in Cancers, Special Issue: MicroRNA-Associated Cancer Metastasis.

Reviewer #4

Comments and Suggestions for Authors:

in the opinion of this reviewer the authors have partially addressed the concerns raised by this reviewer in the first round of review adding in silico analysis but they didn't provide the new experiments required. 

Response:

I appreciate for your detailed review and comments regarding our paper. Your comments will be very useful for our future papers. I performed in silico analyses on this revised MS. Several of your comments are ongoing projects and are subjects for future research. The time to resubmit the paper is limited, please forgive me that it cannot respond to all your comments. Ultimately, I think that judgment of the paper will be left to the Editor's opinion.

Again, I appreciate your review of our paper.

Our data are the first to report these findings in PDAC.

We believe that this article contributes significantly to our understanding of the molecular mechanisms of PDAC pathogenesis. We appreciate your consideration of this manuscript for publication in Cancers, Special Issue: MicroRNA-Associated Cancer Metastasis.

Sincerely,

Naohiko Seki, PhD

Department of Functional Genomics

Graduate School of Medicine, Chiba University

1-8-1 Inohana, Chuo-ku, Chiba 260-8670, Japan
